# Interstitial Photodynamic Therapy of Glioblastomas: A Long-Term Follow-up Analysis of Survival and Volumetric MRI Data

**DOI:** 10.3390/cancers15092603

**Published:** 2023-05-04

**Authors:** Marco Foglar, Maximilian Aumiller, Katja Bochmann, Alexander Buchner, Mohamed El Fahim, Stefanie Quach, Ronald Sroka, Herbert Stepp, Niklas Thon, Robert Forbrig, Adrian Rühm

**Affiliations:** 1Laser-Forschungslabor, LIFE Center, University Hospital, LMU Munich, 81377 Munich, Germany; 2Department of Urology, University Hospital, LMU Munich, 81377 Munich, Germany; 3Max Planck Institute for Psychiatry, Max Planck Society, 80804 Munich, Germany; 4Institute of Neuroradiology, University Hospital, LMU Munich, 81377 Munich, Germany; 5Department of Neurosurgery, University Hospital, LMU Munich, 81377 Munich, Germany

**Keywords:** 5-ALA, glioblastoma, imaging, long term, malignant glioma, MGMT, MRI, photodynamic therapy, recurrence, stereotactic surgery

## Abstract

**Simple Summary:**

Glioblastomas are the most common primary malignant brain tumors, with a devastating survival perspective. The treatment concept of interstitial photodynamic therapy (iPDT) enables the light-induced destruction of tumor cells based on the combination of a photosensitizer that selectively accumulates in the tumor and light to activate the photosensitizer. The tumor region is illuminated by minimally invasively inserted optical fibers. Under this approach, prolonged overall survival was observed. An analysis of the patient characteristics and the evolution of the MRI data before treatment and during follow-up was performed to identify potential predictors of an improved survival outcome. It was found that the methylation status of the DNA-repair enzyme MGMT is an important factor regarding survival. Other commonly assessed parameters, such as the tumor volume, necrosis–tumor ratio, and contrast enhancement after therapy, did not seem to significantly affect survival. Overall, the iPDT-treated patients showed very promising results regarding a sustained absence of their tumors and prolonged overall survival.

**Abstract:**

Background: The treatment of glioblastomas, the most common primary malignant brain tumors, with a devastating survival perspective, remains a major challenge in medicine. Among the recently explored therapeutic approaches, 5-aminolevulinic acid (5-ALA)-mediated interstitial photodynamic therapy (iPDT) has shown promising results. Methods: A total of 16 patients suffering from de novo glioblastomas and undergoing iPDT as their primary treatment were retrospectively analyzed regarding survival and the characteristic tissue regions discernible in the MRI data before treatment and during follow-up. These regions were segmented at different stages and were analyzed, especially regarding their relation to survival. Results: In comparison to the reference cohorts treated with other therapies, the iPDT cohort showed a significantly prolonged progression-free survival (PFS) and overall survival (OS). A total of 10 of 16 patients experienced prolonged OS (≥ 24 months). The dominant prognosis-affecting factor was the MGMT promoter methylation status (methylated: median PFS of 35.7 months and median OS of 43.9 months) (unmethylated: median PFS of 8.3 months and median OS of 15.0 months) (combined: median PFS of 16.4 months and median OS of 28.0 months). Several parameters with a known prognostic relevance to survival after standard treatment were not found to be relevant to this iPDT cohort, such as the necrosis–tumor ratio, tumor volume, and posttreatment contrast enhancement. After iPDT, a characteristic structure (iPDT remnant) appeared in the MRI data in the former tumor area. Conclusions: In this study, iPDT showed its potential as a treatment option for glioblastomas, with a large fraction of patients having prolonged OS. Parameters of prognostic relevance could be derived from the patient characteristics and MRI data, but they may partially need to be interpreted differently compared to the standard of care.

## 1. Introduction

Glioblastoma multiforme (2016 WHO CNS, grade IV) is the most frequent primary malignant neoplasia of the CNS [1]. With its devastating median overall survival (OS) of 14.6 months [2], based on a variety of therapeutic resistances [3,4] and its highly invasive growth [5,6], its treatment remains a formidable challenge in modern medicine.

The current standard of care (SOC) involves maximal safe resection followed by chemoradiation [2,7]. Despite the recent advances in resection with fluorescence-guided surgery [8,9], the prognosis remains daunting. Additionally, patients with tumors located in eloquent or deep-seated areas are often excluded from receiving that form of treatment [10]. To address this issue and to potentially prolong survival while preserving an adequate quality of life, several new approaches were explored based on minimally or noninvasive procedures, such as brachytherapy, tumor-treating fields, radiosurgery, and definitive chemoradiation [10,11,12,13].

Among these approaches, 5-aminolevulinic acid (5-ALA)-mediated interstitial photodynamic therapy (iPDT) was identified as a promising option building on standard stereotactic procedures [14,15,16,17]. Patients receive 5-ALA as a prodrug to achieve the intracellular synthesis of a photosensitizing molecule (protoporphyrin IX, PpIX). Due to an impaired blood–brain barrier within the tumor region and a derailed metabolism in the tumor cells, PpIX accumulates selectively in the malignant cells [18,19,20]. After the minimally invasive placement of optical diffusor fibers, the target tissue is illuminated with laser light. To activate the photosensitizer, light of a wavelength adopted to the absorption characteristic of the photosensitizer has to be applied. In the case of PpIX, a wavelength of 635 nm is recommended [21]. The light-based excitation of PpIX induces the production of reactive oxygen species (especially singlet oxygen), which damages and, finally, kills the neoplastic cells [22]. In comparison with the SOC, no craniotomy is required. Instead, the devitalized tumor mass remains harmlessly in situ. Another advantage is that the healthy tissue is preserved due to the tumor-selective accumulation of PpIX [18]. After iPDT, prolonged median OS (≥24 months) has been observed [14,17,23], although the reasons for this encouraging observation are not yet fully understood.

The effects of iPDT on a tumor and its environment are still under investigation due to the plenitude of involved processes. An immunogenic response may be involved as suggested in a variety of applications of photodynamic therapy (PDT) [24,25,26,27,28,29,30].

To understand the complex processes induced by iPDT, the available data need to be assessed and analyzed systematically. In this study, imaging data, patient characteristics, and molecular biological information were collected and evaluated retrospectively. Due to the fundamentally different principle of iPDT compared to the SOC, quite different findings in the imaging data may be expected. Therefore, an inspection of the peculiarities in the MRI data obtained after iPDT was performed. The characteristic differences from the typical imaging findings after performing conventional therapies were described. To determine potentially survival-related factors, various accessible parameters were examined for correlations with the survival data. With this, the study also strived to explore the potential of iPDT, to motivate further research on this important topic, and to lay a foundation for future investigations.

## 2. Patients and Methods

### 2.1. Patients

The retrospectively analyzed cohort consisted of 16 patients who underwent 5-ALA-mediated iPDT [17]. This procedure was offered to patients suffering from biopsy-proven de novo glioblastomas for whom an SOC treatment was impossible due to the location of the tumor or other contraindications. Instead of definitive chemoradiation [10], iPDT combined with a fractionated 60 Gy radiation therapy and temozolomide (according to Stupp [2]) was offered as salvage treatment [17].

These procedures were performed between 2008 and 2014, providing the opportunity to observe the patients’ course over the long term. This retrospective analysis of the available data was approved by the institutional review board of the Faculty of Medicine at the Ludwig–Maximilians–Universität München, Munich, Germany (ethics approval No. 335–16).

The presented patient cohort features a median age at diagnosis of 65.8 years (range: [29.7; 76.5] years) and a median tumor volume of 6063 mm³ (range: [1362; 21,825] mm^3^). The last follow-up date included in the present analysis was 12 January 2022. At that time, there were three survivors with a median follow-up after diagnosis of 113.9 months (range: [110.3; 127.5] months). All three remained progression-free during that period. Patient information and individual outcomes are summarized in Table 1.

Further clinical aspects regarding these patients can be found in Quach et al. [17]. Details about tumor characteristics and the applied adjuvant therapy are summarized in Table 2.

Glioblastomas show a variety of therapeutic resistances due to their heterogeneous genetic profiles [3,4]. Consequently, new classification criteria were released recently (WHO CNS5) [31]. Since not all tumor characteristics were determined at the time of treatment for the patient cohort reported, the previous classification scheme (2016 CNS WHO) was applied [32], tolerating conceivable changes in classification. The available tumor characteristics of the cohort are shown in Table 2.

Among the regularly recorded MRI scans (in general, every three months), imaging datasets recorded at five specific points of time were selected for this analysis: before treatment, twenty-four hours after treatment, three months after treatment, three months prior to relapse (preceding follow-up, defined retrospectively), and at diagnosis of recurrence. At each instance, the results of three different MRI sequences were analyzed: noncontrast and contrast-enhanced T1-weighted imaging and T2-weighted imaging. Volumes and volume ratios derived from these images are summarized in Appendix A.

### 2.2. Methods

#### 2.2.1. Survival Analysis

Progression-free survival (PFS) was defined as the shortest of the three periods from treatment to radiologically identified progression, palliative discharge, or (censored) end of follow-up. Overall survival (OS) was defined as the shortest of the two periods from diagnosis to death (with or without progression) or (censored) end of follow-up. Postprogression survival (PPS) was defined as the period from diagnosis of progression to death or (censored) end of follow-up.

In the literature, no consistent definition of long-term survival for patients with de novo glioblastomas can be found. Suggested values range from 18 to 60 months [33,34]. In numerous publications, the two-year mark is used as a threshold criterion [35,36,37,38,39]. Based on this, the term “prolonged OS” was decided to be used for patients surviving for at least 24 months after diagnosis (OS) or without progression after iPDT (PFS), respectively. The validity of the chosen term was reaffirmed through a meta-analysis reporting a two-year OS for only 18% (95% CI = [0.14; 0.22]) of the treated glioblastomas [40].

Within the iPDT cohort, the patients’ survival, based on their MGMT-methylation status, was examined. In addition, a comparative evaluation was performed with regard to other patient cohorts who, instead of iPDT, received SOC treatment, a sole chemoradiation treatment, CyberKnife^®^ (Accuray, Sunnyvale, CA, USA) surgery, and stereotactic radiosurgery, respectively [2,10,11,12].

#### 2.2.2. MRI Analysis

Prior to analyzing the images, different preprocessing routines were applied. First, MRI data recorded at different time points were positionally registered to each other. This was performed automatically using the ANTs software package (v2.3.2, University of Pennsylvania, Philadelphia, PA, USA; University of Virginia, Charlottesville, VA, USA; and University of Iowa, Iowa City, IA, USA) [41] followed by manual checks and adjustments. Subsequently, the MRI datasets were semiautomatically segmented using ITK-SNAP (v3.8, University of Pennsylvania, Philadelphia, PA, USA) [42]. All images, adjustments, and segmentations were checked by two experienced neuroradiologists from the Institute of Neuroradiology of the LMU Hospital, Munich (R.F. and K.B.). Volumes of interest are shown exemplarily for one patient (IPDT 14) in Figure 1, such as the tumor volume (consisting of a T1-contrast-enhancing region and a necrotic region), edema, and iPDT remnant. This iPDT remnant appears after treatment and is described and discussed below. For one patient (IPDT 07), the iPDT treatment needed to be performed in two separate sessions, one week apart, due to the tumor’s volume and morphology. Due to this peculiarity, this iPDT treatment did not qualify for the combined analysis of MR images recorded on the first day after treatment. For all other patients, the MR images recorded at this time point could be included in the analysis.

Segmented volumes were rounded to full mm^3^ since a higher accuracy is not meaningful based on the physical imaging resolution. The necrosis–tumor ratio (NTR) was calculated for the pretreatment situation by dividing necrosis volume by tumor volume, corresponding to Henker et al. [43].

#### 2.2.3. Statistics

Information about the segmented volumes, the patients’ individual courses, and clinical parameters was collected in a database. Patients were then divided into subcohorts by recurrence type (local vs. distant) and the methylation status of the MGMT promoter (full/partial methylation vs. no methylation). A double-sided log-rank test was applied to evaluate the statistical significance of differences between each pair of survival curves. Survival analysis was performed using SPSS (v29, IBM Corp., Armonk, NY, USA) [44]. Due to the relatively small number of patients, extensive correlational testing was not feasible. To identify potentially important parameters and relations, the obtained parameters were plotted against survival values using the statistical programming language R (v4.2.1, R Core Team, Vienna, Austria) [45] and the R visualization package GGPLOT2 (v3.3.6, Wickham, 2016) [46]. Data plots that seemed to contain an important message were selected for display and further analysis. Univariate Cox regression analysis was performed to investigate the validity of the visually derived conclusions. Along with the *p*-values, the hazard ratios (HR) are shown in Appendix A.

Since the decease was seen as the most relevant endpoint for the patients, it was decided to restrict the investigations regarding survival-related factors to OS, or PFS in exceptional cases. An analysis primarily focused on PFS and long-term PFS (>24 months) of the iPDT cohort can be found in Quach et al. [17]. For all comparisons, significance level α = 0.05 was appointed.

The survival data of some of the compared patient cohorts had to be extracted from published Kaplan–Meier graphs as numeric data using Engauge Digitizer [47] and had to be subsequently reconstructed into a per-patient format using the programming language R [45]. After a plausibility check, a statistical analysis was performed in comparison to the iPDT cohort. A limitation of this approach was that censored survival data could not be identified and included in survival analysis. Hence, the respective *p*-values should be seen as an estimation. A comparison of the published survival parameters and the corresponding parameters recalculated based on reconstructed data can be found in Appendix A. Besides this limitation, this method enabled a comparison with other patient cohorts for which the detailed original data are unavailable. Whenever original per-patient data of other patient cohorts were available, they were used directly for the comparative analyses.

## 3. Results

### 3.1. Survival

The iPDT-treated cohort showed a median PFS of 16.4 months (95% CI: [5.1; 27.6] months) and a median OS of 28.0 months (95% CI: [6.0; 50.0] months) [17]. The median PPS was 6.3 months (95% CI: [1.6; 11.0] months); see Figure 2 and Table 3. Three patients are still alive (IPDT 03: 128 months) (IPDT 07: 110 months) (IPDT 09: 114 months), and none of them have experienced a recurrence [17]. The survival times (OS, PFS, PPS) obtained with iPDT are compared with those of the patient cohorts treated with other methods in Table 3, and the *p*-values obtained through the pairwise log-rank testing of the differences among some of these survival times are listed in Table 4.

Stratifying the iPDT cohort by the *MGMT promoter methylation status* revealed significantly superior PFS in the case of a methylated MGMT promoter (median of methylated MGMT of 35.7 months vs. median of unmethylated MGMT of 8.3 months; *p* = 0.030). Similarly, OS was significantly longer in the case of a methylated MGMT promoter (median of methylated MGMT of 43.9 months vs. median of unmethylated MGMT of 15.0 months; *p* = 0.031). PPS appeared to be more favorable for the methylated group, but the difference was not statistically significant according to log-rank analysis (median of methylated MGMT of 15.3 months vs. median of unmethylated MGMT of 3.8 months; *p* = 0.192). Through a univariate Cox regression analysis, as in Quach et al. [17], the MGMT status was found to be a potentially survival-related parameter with a significant reduction in the risk of death for patients with a methylated MGMT promoter (*p* = 0.042; hazard ratio (HR) = 0.280; 95% CI of HR = [0.082; 0.956]).

When comparing the entire iPDT cohort with the *standard-of-care (SOC)* cohort according to *Stupp* et al. (*n* = 287) [2], the iPDT cohort showed superior PFS (median of iPDT of 16.4 months vs. median of Stupp of 6.9 months; *p* = < 0.001) and OS (median of iPDT of 28.0 months vs. median of Stupp of 14.6 months; *p* = 0.017). For the Kaplan–Meier plots, see Appendix A.

A comparison of iPDT with a *sole chemoradiation treatment* (*n* = 56) [10] also revealed superior PFS (median of iPDT of 16.4 vs. median of chemoradiation of 8.0 months; *p* = 0.005) and OS (median of iPDT of 28.0 months vs. median of chemoradiation of 12.0 months; *p* = 0.022) for the iPDT cohort. No significant difference in the PPS values was found (median of iPDT of 6.3 months vs. median of chemoradiation of 4.0 months; *p* = 0.516).

In comparison with the *CyberKnife^®^ surgery* (*n* = 12) [12], no significant differences could be discovered (median PFS of iPDT of 16.4 months vs. median PFS of CyberKnife^®^ surgery of 16.0 months; *p* = 0.341) (median OS of iPDT of 28.0 months vs. median OS of CyberKnife^®^ surgery of 18.0 months; *p* = 0.280) (median PPS of iPDT of 6.3 months vs. median PPS of CyberKnife^®^ surgery of 3.0 months; *p* = 0.237).

Lastly, in comparison with the *stereotactic radiosurgery* (*n* = 30) [11], iPDT featured superior OS (median of iPDT of 28.0 months vs. median of stereotactic radiosurgery of 14.8 months; *p* = 0.036).

Side effects mainly consisted of transient health restrictions such as aphasia. In one case, a pulmonary embolism was seen [17,48]. However, it is important to note that iPDT is not necessarily accountable for the embolism. Brain tumor patients generally bear significant risk factors for thromboembolic events, such as the preexisting malignancy itself, often having an advanced age, and immobilization due to the disease or treatment [49].

### 3.2. Volumetric Assessments

The time courses of contrast-enhanced T1-weighted MRIs are exemplarily shown for two patients in Figure 3. Both patients experienced prolonged OS and also a PFS of more than 24 months. The first patient (IPDT 02, PFS of 59.2 months, OS of 95.0 months) exhibited a tumor volume of 1362 mm^3^. Before treatment, the tumor comprised a T1 contrast enhancement (CE) volume of 878 mm^3^ and a necrosis volume of 484 mm^3^. At 24 h after iPDT, a small CE volume (22 mm^3^) remained. Three months after iPDT, no evidence of CE was found. The second patient (IPDT 14, PFS of 35.7 months, OS of 43.9 months) featured a tumor volume of 15,335 mm^3^ (CE volume of 8929 mm^3^ and necrosis volume of 6406 mm^3^). At 24 h after iPDT, a CE volume of 4720 mm^3^ remained. Three months after iPDT, no evidence of CE was found in this patient.

The statistical information (median, average, minimum, maximum, and standard deviation) on the volumes of these and other distinguishable tissue regions is collected in Table 5. The individual values for each patient are provided in Appendix A.

The median tumor volume to be treated was 6063 mm^3^ (range: [1362; 21,825] mm^3^), consisting of a contrast-enhancing region (median volume of 3533 mm^3^; range of [878; 12,362] mm^3^) and a necrotic region (median volume of 1350 mm^3^; range of [118; 9463] mm^3^). At 24 h after iPDT, 13 patients showed residual CE. Collectively, over all 15 evaluable cases, the median volume of residual CE was 535 mm^3^ (range: [0; 4720] mm^3^). In seven patients, residual CE at the former tumor location was still present after three months.

In the MR images recorded 24 h after iPDT, morphological changes appeared in the native T1- and T2-weighted images around the former tumor location (labelled as volume “4” in Figure 1). The affected region (presenting as circumscribed predominantly T1 hypointensity and T2 hyperintensity) was defined as an “iPDT remnant”, representing the devitalized tumor residing in situ. This region featured a median volume of 12,426 m^3^ (range: [2062; 66,857] mm^3^). In 14 patients, the iPDT remnant was larger than the original tumor; in 1 patient, the remnant was smaller.

At the first follow-up (typically 3 months after iPDT), the median iPDT remnant volume decreased to 5585 mm^3^ (range: [370; 20,027] mm^3^). In four cases, the remnant remained larger than the original tumor. In eight cases, the remnant shrank to a volume smaller than the original tumor. In the remaining four cases, no MRI data were available at this point of time. Among the 12 patients who experienced a recurrence, 3 months prior to the relapse, the median iPDT remnant volume decreased even further to 4036 mm^3^ (range: [305; 12,254] mm^3^). In 2 of these 12 cases, the remnant was larger than the original tumor; in 7 cases, the remnant was smaller. The median iPDT remnant volume at the time of recurrence amounted to 3914 mm^3^ (range: [157; 9607] mm^3^). In three cases, the iPDT remnant at this time point was found to be bigger than the original tumor; in the remaining nine cases, the remnant was smaller.

The median volume of the 12 recurrent tumors was 5893 mm^3^ (range: [6; 17,177] mm^3^), consisting of 8 local and 4 distant relapses. The distant recurrences had a median distance of 29.5 mm to the margin of the iPDT-treated tumor (IPDT 06: 39 mm) (IPDT 08: 31 mm) (IPDT 13: 26 mm) (IPDT 14: 28 mm) (values rounded).

### 3.3. Relations between Observations

In Figure 4, the selected graphs are displayed to illustrate the possible relations between the different parameters and observations. For further information, see Table 1 and Table 2 and Appendix A (for univariate Cox regression results, see Appendix A).

Figure 4a (for numeric values, see Table 1) displays the *age at diagnosis* vs. *OS*. Prolonged OS was seen in all the patients with an age under 60. For patients older than 60 years, three of nine cases still showed prolonged OS. (*p* = 0.067; HR = 1.038; 95% CI = [0.997; 1.080].) Although slightly missing significance, the resulting hazard ratio still suggests a worse OS prognosis for patients with a higher age at diagnosis.

Figure 4b (for numeric data, see Table 2) displays the *number of TMZ cycles after chemoradiation* vs. *OS*. OS below 24 months was only seen in the case of two or less TMZ cycles (typical cycle period of 1 month). Within this subgroup of two or less cycles, three of eight cases still showed prolonged OS. All the other patients showed prolonged OS (*p* = 0.146; HR = 0.859; 95% CI = [0.700; 1.055]).

Figure 4c (for numeric values, see Appendix A) displays the *necrosis–tumor ratio (NTR)* vs. *OS*. No pattern indicating a systematic relation could be derived in this case (*p* = 0.540; HR = 0.279; 95% CI = [0.005; 16.533]). The high *p*-value and the broad 95% CI also suggest that this cohort shows no correlation between the NTR and OS.

Figure 4d (for numeric values, see Table 1) displays the *PPS* vs. *PFS* values. It was observed that longer PFS was accompanied by longer PPS and vice versa (*p* = 0.053; HR = 0.938; 95% CI = [0.879; 1.001]). It is to be noted that only four of the seven patients with PFS > 24 months are included in this graph because three of them have not experienced progression. No further systematic trends could be found.

Six examples of tumor-related volumes lacking a clear trend in relation to OS are shown in Figure 5 (see also Appendix A), namely for the *tumor volume* (Figure 5a; *p* = 0.653; HR = 1.020; 95% CI = [0.935; 1.112]) and the *contrast-enhancing volume 24 h after iPDT* (Figure 5b; *p* = 0.599; HR = 1.096; 95% CI = [0.779; 1.541]). In addition, the volumes of the *iPDT remnant* are shown at the time points 24 h after iPDT (Figure 5c; *p* = 0.401; HR = 0.986; 95% CI = [0.954; 1.019]), 3 months after iPDT (Figure 5d; *p* = 0.341; HR = 0.946; 95% CI = [0.845; 1.060]), 3 months before recurrence (Figure 5e; *p* = 0.051; HR = 1.318; 95% CI = [0.998; 1.740]), and at recurrence (Figure 5f; *p* = 0.156; HR = 1.144; 95% CI = [0.950; 1.377]).

In Figure 6 (see also Appendix A), the distribution of the PFS and OS values is shown, stratified by MGMT promoter methylation and the presence of CE three months after iPDT. No correlation between the presence of CE and survival (PFS and OS) could be discerned (e.g., with regard to OS, *p* = 0.223, HR = 2.259, and 95% CI = [0.610; 8.363]). Additionally, the MGMT methylation status did not show any correlation with CE three months after iPDT.

## 4. Discussion

### 4.1. Survival Comparison

Compared to the cohort of Stupp et al. [2], the iPDT cohort exhibited significantly longer median PFS and OS. This comparison to a prospective study with randomized arms must be interpreted cautiously due to the following limitations: The control of potential confounders was not sufficient in the iPDT cohort (e.g., age, symptomatology, the size of the tumor, and the impact of other molecular biological markers). Treated individuals were not selected randomly, and the overall number of treatments was small. Nonetheless, this comparison can help to estimate the survival outcome of iPDT treatments and motivates further investigations due to these promising results.

In comparison to the *chemoradiation-treated cohort* [10], the iPDT cohort also showed promising results. Again, these results must be interpreted cautiously, as the iPDT study lacked a prospective design. The patients were treated with iPDT due to the nonresectability of their tumors. The promising results after iPDT in this rather desperate clinical situation provide an outlook on the potential of iPDT treatments.

In comparison with *stereotactic radiosurgery* [11], prolonged OS could be confirmed for the iPDT cohort, outliving the SOC comparison cohort by an additional 13.2 months (based on the median OS values). This comparison can be seen as quite meaningful and relevant since both cohorts represent highly selected groups with a small number of cases. Still, the differences in the patient characteristics are apparent and need to be addressed in further studies.

Compared to the *CyberKnife*^®^ treatment, the iPDT cohort outlived the SOC comparison cohort by an additional 10 months (based on the median OS values), although this result did not reach statistical significance. These cohorts did match well since a good overall status, a lack of randomization, and a small number of patients (*n* = 12) were shared by both cohorts.

Overall, it can be concluded that this specific iPDT cohort showed promising survival. This is a favorable fact which is highly worthwhile to be corroborated in further clinical trials such as NCT03897491. In further investigations, other minimally invasive therapy approaches should also be included in the survival comparison. Examples are seed-based brachytherapy [13,50] and laser-induced thermal therapy [51,52]. In contrast to iPDT, these therapies do not comprise tumor selectivity but rely on the optimal tailoring of the therapeutic effect to a predefined treatment volume that is to be destroyed entirely [13,50,51,52].

The *MGMT status* played a significant role in the treatment outcome of the iPDT cohort [17]. The iPDT patients with a methylated MGMT promoter outlived those with an unmethylated MGMT promoter by an additional 28.9 months (based on the median OS values), and they remained progression-free for an additional 27.4 months (based on the median PFS values).

Such survival advantages in the case of a methylated MGMT promoter have been reported for many other treatment approaches [10,53,54,55,56]. This is commonly attributed to a higher sensitivity of the tumor cells to adjuvant chemotherapy with temozolomide [54,57]. In the case of iPDT-treated recurrent glioblastomas, no survival benefit was found for a methylated MGMT promoter compared to an unmethylated MGMT promoter [16]. This might be due to a higher resistance of these patients’ tumors to chemotherapy due to the inhibition of apoptosis [58] or an upregulation of the genes causing multidrug resistance [59,60]. Still, the cohort with recurrent glioblastomas showed promising survival [16]. Additionally, in comparison to the patient cohort suffering from newly diagnosed glioblastomas and being treated with chemoradiation as reported by Hegi et al. [56] (see Table 6), the iPDT cohort reported here showed superior survival both for methylated and unmethylated MGMT promoters. In conclusion, it seems that iPDT shows promising survival with an even stronger improvement in prognosis when the tumor has a higher sensitivity to temozolomide.

### 4.2. Posttreatment CE

For the evaluation of the standard therapies, the course of posttreatment CE was an important aspect [61,62]. First, the observation that, 24 h after iPDT, CE was apparent in some patients is discussed with regard to its impact on the survival of the iPDT-treated patients and in relation to the common findings after the SOC.

After the SOC, the residual CE in the MRI data recorded within the first 72 h after open resection is associated with remaining vital tumor tissue [63]. This indicates an incomplete resection, suggesting that the complete resection of contrast-enhancing tumors (CRET) failed, which is associated with a significant reduction in OS [64,65,66,67,68]. Likewise, in case of iPDT, the residual CE in the early imaging after iPDT may, at first sight, be interpreted as an incomplete response, which, in turn, is generally considered as a negative predictive factor for survival [23]. However, no correlation of this observation in the MRI data with the therapy outcome could be derived from the present study (see Figure 5b), which is contrary to the respective statements in the review by Leroy et al. [23]. In the iPDT patient cohort of the present study, residual CE less than 72 h after iPDT was not significantly associated with inferior survival. This suggests that residual CE may not necessarily indicate residual vital tumor tissue but rather some remaining disturbance of the vasculature and the blood–brain barrier (BBB) after iPDT. This assumption relies on the core principle of the contrast agent that various conditions (e.g., tumors, ischemia, and autoimmune diseases) affect the BBB, thus allowing the agent to emerge [69,70]. In the case of residual CE after iPDT, it might indicate a remaining local alteration of the BBB in the treated tissue volume. This hypothesis is affirmed by the knowledge that 5-ALA mediated PDT has the potential to temporarily disrupt the BBB [71]. This may possibly help to enable immunogenic effects. The general role of the involvement of the immune system is known from other applications of the PDT [24,25,26,27,28,29,30]. The affected BBB might support or even enable the immune system to react on the glioblastoma tissue. A detailed understanding of this complex topic requires further investigation.

In any case, it must be concluded that the presence of CE 24 h after treatment is not well suited as a predictor of the completeness of response in the case of iPDT, at least not in the case of this particular iPDT-treated patient cohort. This can be seen as a paradigm shift for physicians and radiologists since a formerly clearly negative predictor is now to be interpreted more carefully.

Three months after iPDT, CE was found in seven patients around the former location of the tumor. Generally, CE beyond the 72-h threshold can either be associated with a reaction to the treatment or can indicate residing/progressive tumor tissue. This complicates the differentiation between a therapy reaction and tumor progression by means of conventional MRI sequences [63]. Similarly, in 20% to 30% of the SOC-treated patients, small changes in the contrast-enhancing regions or the edema were observed within the first 3 to 6 months after treatment [72]. This phenomenon of pseudoprogression is supposed to be based on postsurgical changes, radiation necrosis [73], and/or inflammation [74]. To differentiate these changes from a progressive disease, response assessment criteria for high-grade gliomas were developed based on the SOC data [61,62]. Since the role of CE after iPDT remains uncertain, an adjustment of the evaluation criteria for the case of iPDT should be pursued.

The role of residual CE after iPDT should also be examined regarding the supposed immunogenic response as observed in other applications of PDT [24,25,26,27,28,29,30].

### 4.3. iPDT Remnant

The composition of the iPDT remnant will remain unclear as long as the systematic analyses of posttreatment biopsies are lacking. Generally, it was found that the PDT treatment induces a mixture of apoptosis and necrosis [29,30,75,76,77]. Consequently, signatures of both processes may be expected in the iPDT remnant region. Investigating the iPDT remnant for apoptosis- and necrosis-related effects would be desirable for an enhanced understanding of the involved processes.

### 4.4. Recurrence Patterns

According to the literature, 90–95% of all glioblastomas reoccur within 2 cm from the primary tumor’s margin as observed in a plenitude of other glioblastoma cohorts [78,79,80,81,82]. The remaining 5–10% of recurrences outside of that 2 cm margin may be termed distant recurrences. The high number of local recurrences might be due to insufficient local tumor cell eradication. One explanation could be that the tumor margins apparent in conventional MRI do not include the tumor cells spread into the local environment [83]. In the analyzed iPDT-treated cohort, 4 of the 12 recurrences (33%) were distant recurrences according to the criterion mentioned above. It may be concluded that a smaller fraction of local recurrences is observed after iPDT compared to other treatments. This suggests a superior sustained local tumor control (the permanent absence of vital tumor cells within the former tumor location and its surroundings) compared to other approaches. No correlation between the distance of the recurrence from the primary tumor and any other evaluated parameter could be found in this study. Potential reasons would need to be further investigated, for example, a better (or even complete) eradication of tumor cells due to the induction of an immunogenic response [24,25,26,27,28,29,30]. Besides the involvement of the immune system, there might be another explanation: Due to the oral application of 5-ALA, the photosensitizer PpIX could accumulate in all the tumor cells regardless of their location [18]. An uptake of the photosensitizer outside of the CE-based tumor margins, indicated by photosensitizer-specific fluorescence, was also observed in fluorescence-guided resection [84]. This means that the tumor cells in the surrounding of the MRI-based tumor volume also accumulated the photosensitizer, which made them vulnerable to illumination, while the physiological cells were preserved [22]. Although the used target fluence (18.72 J/mm^2^ [15]) may not be reached in that tissue region, the photosensitizer may still become activated and cause cell damage and/or death to some extent. These two aspects may mutually reinforce each other. A detailed assessment and analysis of the immunogenic response to iPDT would be desirable, especially in combination with examining other parameters, e.g., the supposedly compromised BBB (see Section 4.2).

### 4.5. Examination of Possibly Survival-Related Factors

In the case of resection, the following aspects are considered to be of prognostic importance: the patient’s age, tumor volume, extent or completeness of tumor resection, degree of necrosis, involvement of the eloquent cortex or a deep structure, and pretreatment uptake of the contrast agent [68,85,86]. Additionally, the necrosis–tumor ratio (NTR) was also regarded as relevant for the prognosis [43].

For iPDT, a preoperative Karnofsky performance scale (KPS) score over 70, a well circumscribed roughly spherical lesion, and strong PpIX accumulation were seen as favorable factors [23]. Additionally, a complete response according to early posttreatment brain imaging (a major reduction in CE 24 h after iPDT) and a small tumor volume before treatment (<5000 mm^3^) were reported to be beneficial to survival [23].

In the analyzed iPDT cohort, it was seen in Figure 4a that *a younger age at diagnosis* (<60 years) led to prolonged OS in all cases. This suggests that a younger age is a favorable predictive factor, which is consistent with the equivalent findings of the SOC [68,85,86]. Nonetheless, 3/9 patients (33%) with an age over 60 years experienced long-term survival. It can be concluded that an age < 60 is a positive predictive factor, but a higher age still does not rule out a prolonged OS. Thus, iPDT seems to provide a favorable survival perspective for all age groups.

Concerning the number of *TMZ cycles* after iPDT (see Figure 4b), a higher number of cycles also seems to have a positive influence on the outcome. Here, longer OS might be confounded by an otherwise good health status and a correspondingly better toleration for chemotherapy. However, prolonged survival was also seen in combination with a low number of TMZ cycles, affirming considerable tumor eradication with iPDT by itself.

Surprisingly, the *necrosis–tumor ratio (NTR)* (see Figure 4c) does not seem to have any influence on the outcome. This observation is contrary to the SOC, where an NTR above 0.33 is seen as a negative predictor regarding OS [43]. As an explanation, a higher NTR is supposed to be associated with a more invasive and more resistant type of glioblastoma [43]. However, the NTR does not seem to be relevant to iPDT, at least for the present patient cohort. This, again, suggests good local tumor control with iPDT, even in the case of more invasive tumors.

Patients with long *progression-free survival (PFS)* (see Figure 4d) also tended to show long *postprogression survival (PPS)*, which might be explainable by a better overall health state and/or good tumor control in the respective cases.

The *tumor volume* (Figure 5a) showed no correlation with OS. This is in contradiction to common experiences in the treatment of glioblastomas. However, it has to be kept in mind that the tumor diameter was limited to 4 cm by the inclusion criteria for the iPDT cohort (see Quach et al. [17]). Specifically, a tumor volume < 5000 mm^3^ was postulated to be a favorable factor for prolonged OS after iPDT [23]. When comparing this value to the median tumor volume of the present iPDT cohort of 6063 mm^3^ (range: [1362; 21,825] mm^3^), the suggested threshold was clearly exceeded. Nevertheless, a large fraction of the patients showed prolonged OS (10/16 patients overall (62.5%) and 6/9 patients with a tumor volume > 5000 mm^3^ (67%). The postulated threshold tumor volume of 5000 mm^3^ was apparently based on the research of Kaneko et al. [87]. This work did not involve dosimetry calculations, so an incomplete illumination of the tumor tissue cannot be ruled out. As a consequence, the iPDT effects may have been triggered insufficiently, particularly in the case of larger tumors. Hence, a nonnegligible number of vital tumor cells may have survived the therapeutic intervention, resulting in a situation comparable to an incomplete resection in the SOC. Therefore, it is crucial that the iPDT procedure is well planned to properly illuminate the target volume [88] since iPDT relies on the interaction of light with the photosensitizing drug [89]. The obtained results suggest that thorough treatment planning allows one to avoid a deterioration of iPDT’s success with increasing tumor volume [90,91]. Additionally, advances in dosimetry planning [92,93,94] and intraoperative monitoring [21] should make it possible to treat larger tumors in the future.

Like the primary tumor volume, the *volume of the iPDT remnant* (Figure 5c–f) also did not show any correlation with survival. Despite the low *p*-value in the univariate Cox regression analysis three months before recurrence, a causal relationship with survival in this case seems unlikely. Considering that no significant correlation of OS was found with respect to the iPDT remnant volume shortly after iPDT treatment and three months after iPDT treatment, the low *p*-value might be explained by the time-dependent decrease in the iPDT remnant: the longer overall survival is, the longer the iPDT remnant volume can decrease. This would result in a smaller remnant if recurrence occurred after a long survival period. This implies a correlation but not causality.

Furthermore, no connection with OS was found for the CE volume 24 h after iPDT (Figure 5b) or for the observation of CE 3 months after iPDT (Figure 6).

All in all, these results for iPDT differ in several ways from the common experience in resection, where the complete resection of the contrast-enhancing tumor volume (i.e., vanished CE in postoperative imaging) is a well-accepted predictor of prolonged OS [68,86,95]. In conclusion, the reactions to the iPDT intervention seem to be different from the reactions to the other therapy approaches.

### 4.6. Quality Assessment of the Performed Treatment

It appears trivial that the quality of the iPDT treatment must have a crucial influence on survival. From this study, no specific quality criterium can be derived. Further research is needed to define specific criteria for a deeper evaluation in that respect. For now, spectral online monitoring allows for the measuring of PpIX-fluorescence within the tumor volume to confirm the availability of the photosensitizer at the start of the iPDT-illumination process [21,96].

### 4.7. Discussion of Materials and Methods

When critically evaluating the used materials and methods, establishing an artificial-intelligence-based segmentation method would be desirable to facilitate the evaluation of larger patient cohorts in future studies. Additionally, a larger patient cohort would have been helpful for testing the statistical significance of the more survival-related factors. As further means to evaluate and potentially improve the success of iPDT, advanced MRI techniques (e.g., diffusion-weighted imaging (DWI), MR perfusion imaging, and MR spectroscopy) as well as positron emission tomography (PET) imaging may be helpful, as they allow one to assess tumor spreading and to identify the regions associated with a high risk of tumor recurrence and progression [97].

### 4.8. Limitations

Although the median tumor volume of the iPDT cohort was comparatively small, this does not necessarily reduce the relevance of the general success of the performed iPDT treatments. It is to be noted that the presented iPDT cohort was a highly-selected cohort, especially in comparison to the patient cohorts treated according to the Stupp protocol [2] or with chemoradiation [10]. Nonetheless, without iPDT, the treated individuals, with their nonresectable tumors, would have had a daunting prognosis [10,85,86]. With iPDT, 10/16 patients (62.5%) experienced prolonged OS, suggesting that the treated individuals strongly benefitted from the offered salvage treatment.

Despite the truly encouraging outcome in this small patient cohort, the potential pitfalls of iPDT in general might contribute to the fact that early recurrence was still observed, especially in the cases with an unmethylated MGMT promoter. PDT induces hypoxia, which stimulates the proliferation of glioblastoma stem cells [98] and might activate angiogenic factors [99]. It is well established that PpIX is a substrate for some of the multidrug-resistance-associated membrane transporters, such as ABCB1 [100] or ABCG2 [101,102]. As the expression of these transporters is associated with the stemness of cells, PDT might, therefore, lead to the preferential survival of glioblastoma stem cells, although it has been shown that glioblastoma stem cells are susceptible to 5-ALA based treatment [103]. Nitric oxide (NO) production was observed in glioma cells treated with a sublethal dose of 5-ALA-PDT [104], which led to increased proliferation and migration. As far as resistance to repetitive PDT is concerned, the reports are controversial: while Madson et al. [105] did not find a buildup of 5-ALA-PDT resistance in glioma spheroids, PDT-resistant cells could be induced with multiple PDT treatments as found by Casas et al. [106]. More caveats to consider in GBM-PDT have recently been reviewed by Miretti et al. [107]. It appears worthwhile to investigate which of the above-mentioned potential restrictions are clinically relevant in order to develop targeted mitigation strategies that might improve the clinical outcome of 5-ALA based iPDT even further.

This investigation was intentionally focused on volumetric MRI analyses during longtime follow-ups. Priority was given to the clinically relevant *Response Assessment in Neuro-Oncology* (RANO) criteria, such as the contrast-enhancing volume and necrosis post-iPDT [97]. Advanced MRI techniques (e.g., quantitative analyses of contrast-enhanced T1 signal intensities and diffusion-weighted imaging (DWI) with a quantitative evaluation of the apparent diffusion coefficient) were not part of the present nonexhaustive analysis but might depict significant correlations with the overall outcome or at least with local tumor control [108].

### 4.9. Recapitulation and Outlook

The presented data indicate that the first results obtained through the iPDT-treatment of patients with de novo glioblastomas are promising, e.g., in terms of OS. The possibility of treating nonresectable tumors should especially be highly appreciated, as it would greatly improve the prognosis of the affected patients. For recurrent glioblastomas, iPDT was already tested on a larger number of patients, showing the feasibility and safety of the treatment [16].It appears that the good performance of iPDT regarding survival is not only due to a good local tumor-debulking effect but is also based on immunogenic effects. This can be concluded from the observed unusual recurrence pattern characterized by a lower rate of local recurrence than that seen in a plenitude of other glioblastoma patient cohorts [78,79,80,81,82].The confirmation of the importance of the MGMT methylation status [17] can be seen as an important step in the improvement of the patient selection of iPDT for de novo glioblastomas.The assessment of MRI data showed peculiarities compared to the SOC treatment. In the case of residual or increasing CE after iPDT, advanced imaging techniques (e.g., DWI, MR perfusion and spectroscopy, PET) should be considered to enable better differentiation between vital tumor reactions and tissue reactions/pseudoprogression.

In the future, iPDT could also be combined with other therapies to eliminate as many neoplastic cells as possible and to initiate a broad variety of processes that can help to combat this invasive malignancy. To improve the distant tumor control, a combination with a sonodynamic approach might be interesting to better eliminate disseminated tumor cells [109,110,111,112,113].

In glioblastomas, the first trials of the therapies with immune checkpoint inhibitors did not show very encouraging outcomes [114,115]. However, in the other applications, a combination of 5-ALA-mediated PDT with immune checkpoint inhibitors revealed promising results [116]. This could lead to an enhanced response of glioblastoma cells to these new treatment drugs. Likewise, the variety of glioblastomas’ therapeutic resistances may be addressable by combining the current iPDT protocol with chemoradiation-enhancing substances [117]. These approaches are to be tested in future studies.

To improve the perspectives for glioblastoma patients, a large repertoire of therapies should be provided against the burden of glioblastomas in the future. This study explored the potential of iPDT to contribute to the fight against this deadly disease and to provide motivation for scientists and physicians all over the world to further investigate this promising therapy option.

## 5. Conclusions

In this study, iPDT has shown its potential based on a large fraction of patients (63%) experiencing prolonged OS, although this patient cohort was faced with an a priori very daunting prognosis due to the nonresectability of their tumors. In these individuals, the treatment was capable of debulking the tumor burden while only causing transient morbidity.

Furthermore, this investigation demonstrated that survival-related parameters could be derived from commonly available patient data. However, it is to be noted that some of these parameters may need to be interpreted differently compared to the standard treatment. Likewise, a paradigm shift seems to be required with regard to the established interpretation of imaging data since the known predictive factors suitable for other treatments do not seem to be directly transferable to the case of iPDT.

In conclusion, iPDT presents promising survival perspectives and broad applicability, even in the case of problematic tumor locations. With its minimally invasive nature, iPDT carries a notable potential to contribute to the combat against glioblastomas. This potential should certainly be explored further.

## Figures and Tables

**Figure 1 cancers-15-02603-f001:**
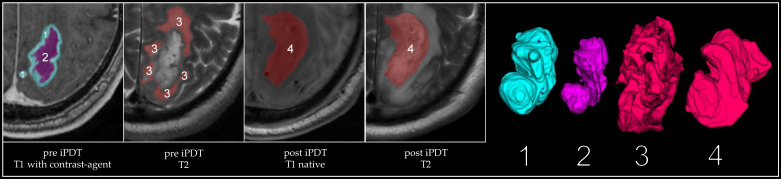
Segmented volumes indicated by color overlays in MR images (T1 and T2 sequences), and separate images indicate 3D volume (1: T1-contrast-enhancing volume) (2: necrotic volume) (1 + 2: tumor volume) (3: edema) (4: iPDT remnant), exemplarily shown for one patient (IPDT 14).

**Figure 2 cancers-15-02603-f002:**
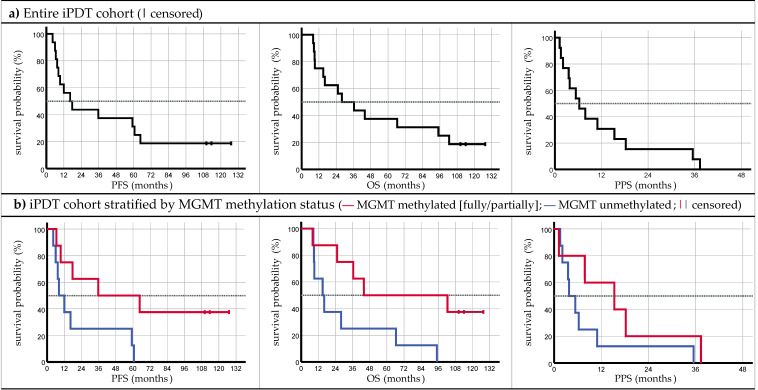
Kaplan–Meier plots related to PFS, OS, and PPS of the (**a**) iPDT cohort as a whole and (**b**) stratified by MGMT methylation status.

**Figure 3 cancers-15-02603-f003:**
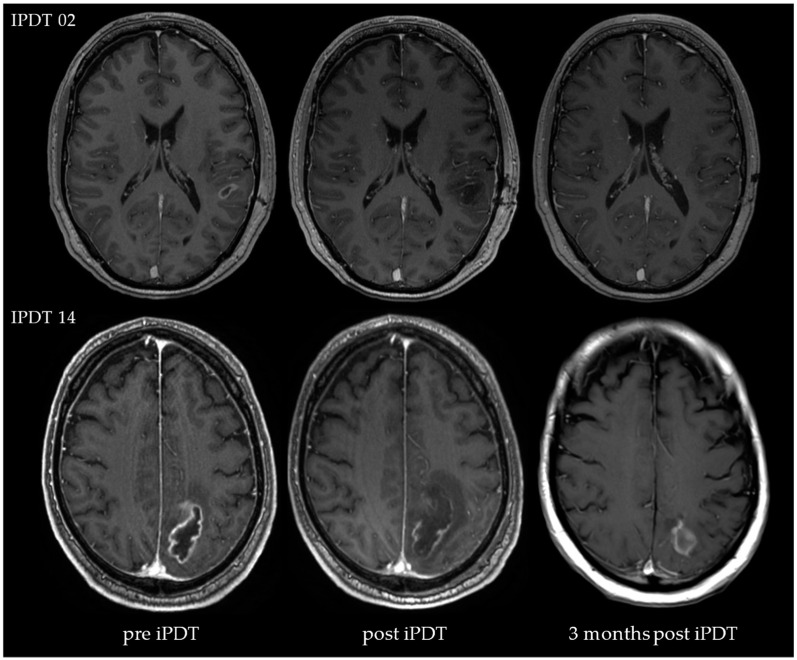
Two exemplary courses of MR images (T1-weighted with contrast agent) recorded before iPDT, twenty-four hours after iPDT, and three months after iPDT (patient IPDT 02 and IPDT 14).

**Figure 4 cancers-15-02603-f004:**
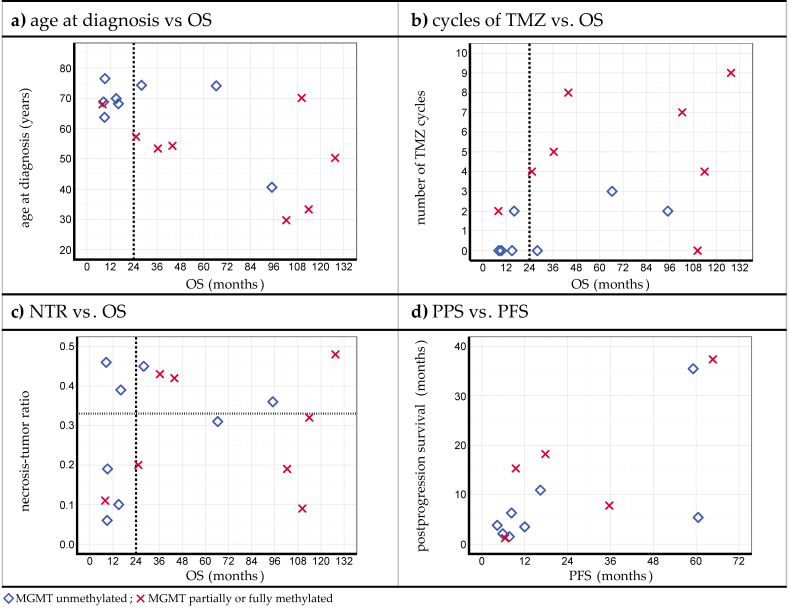
Plots of several parameters for the iPDT cohort vs. OS, (**a**) age at diagnosis, (**b**) number of TMZ cycles, and (**c**) necrosis–tumor ratio (NTR), and vs. PFS, (**d**) postprogression survival (PPS). The dashed vertical line in panels (**a**–**c**) marks the 24-month threshold value introduced to define prolonged OS. The dotted horizontal line in panel (**c**) represents the lower cutoff NTR value associated with a detrimental influence on survival in standard of care (SOC) as suggested by Henker et al. [43].

**Figure 5 cancers-15-02603-f005:**
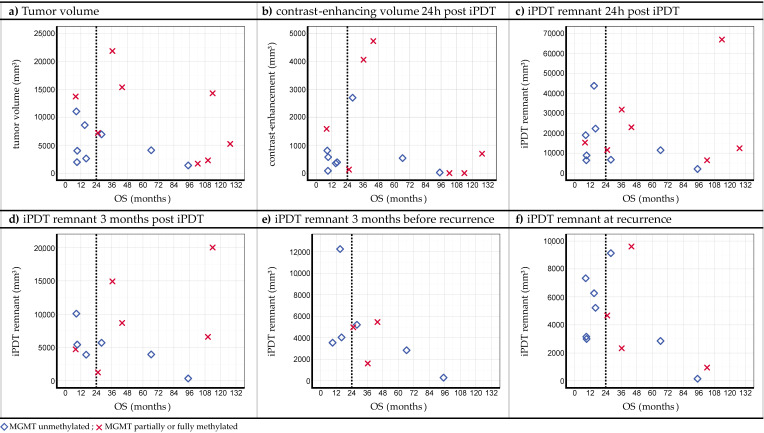
Plots of segmented volumes vs. OS, (**a**) tumor volume, (**b**) contrast-enhancing volume, and (**c**–**f**) iPDT remnant volume, determined at four different time points. The dashed vertical line marks the 24-month threshold value introduced to define prolonged OS.

**Figure 6 cancers-15-02603-f006:**
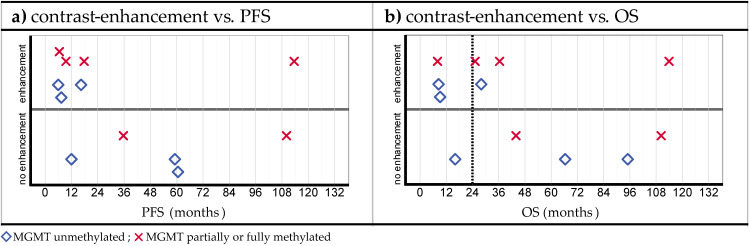
(**a**) Progression-free survival (PFS) and (**b**) overall survival (OS) stratified by the presence or absence of contrast enhancement 3 months after iPDT.

**Table 1 cancers-15-02603-t001:** Patient characteristics and individual outcomes (values rounded, contains selected and, on the other hand, partially extended information in comparison to Table 1 of Quach et al. [17]).

IPDT No. (⟳ PFS, † OS)	Sex (m/f)	Age at PDT (Years)	PFS (Months)	OS (Months)	PPS (Months)	Status
**IPDT 01** (⟳ 65, † 102)	m	29.7	64.7	102.4	37.4	†
**IPDT 02** (⟳ 59, † 95)	m	40.6	59.2	95.0	35.5	†
**IPDT 03** (128)	f	50.3	127.1 *	127.5 *	-	A
**IPDT 04** (⟳ 8, † 15)	m	69.9	8.3	15.0	6.3	†
**IPDT 05** (⟳ 12, † 16)	m	68.2	12.0	16.1	3.5	†
**IPDT 06** (⟳ 4, † 9)	m	63.7	4.3	9.0	3.8	†
**IPDT 07** (110)	m	70.1	110.1 *	110.3 *	-	A
**IPDT 08** (⟳ 61, † 66)	f	74.1	60.6	66.4	5.4	†
**IPDT 09** (114)	m	33.3	113.6 *	113.9 *	-	A
**IPDT 10** (⟳ 16, † 28)	f	74.3	16.4	28.0	10.9	†
**IPDT 11** (⟳ 6, † 9)	m	68.8	6.0	8.5	2.1	†
**IPDT 12** (⟳ 7, † 8)	m	68.0	6.5	8.0	1.2	†\REC
**IPDT 13** (⟳ 10, † 25)	f	57.3	9.5	25.2	15.3	†
**IPDT 14** (⟳ 36, † 44)	m	54.3	35.7	43.9	7.8	†
**IPDT 15** (⟳ 7, † 9)	m	76.5	7.4	9.2	1.5	†
**IPDT 16** (⟳ 18, † 36)	m	53.4	17.8	36.4	18.2	†
**Median**		65.8	16.4 ^a^	28.0 ^a^	6.3 ^a^	
**Average**		59.5	43.1 ^a^	52.9 ^a^	11.5 ^a^	
**Max**		76.5	127.1	127.5	37.4	
**Min**		29.7	4.3	8.0	1.2	

PFS = ⟳ = progression-free survival, OS = † = overall survival, PPS = postprogression survival, m = male, f = female, † = deceased with recurrence, †\REC = deceased without recurrence (palliative discharge), A = alive, * = last follow-up (survivors are still progression-free), and ^a^ = value calculated through Kaplan–Meier survival analysis in SPSS.

**Table 2 cancers-15-02603-t002:** Tumor characteristics and additional therapy regimes (contains selected and, on the other hand, extended information in comparison to Table 1 of Quach et al. [17]).

IPDT No. (⟳ PFS, † OS)	MGMT Methylation ^b^	IDH1	IDH2	Ki67	Tumor location	Side	TMZ during Radiation	TMZ Cycles after Radiation
**IPDT 01** (⟳ 65, † 102)	yes	yes	no	n/a	frontal supraventricular	right	yes	7
**IPDT 02** (⟳ 59, † 95)	no	no	no	10%	temporoparietal	left	partially	2
**IPDT 03** (128)	yes	no	no	10%	temporo-occipital	left	yes	9
**IPDT 04** (⟳ 8, † 15)	no	no	no	30%	temporal	left	yes	0
**IPDT 05** (⟳ 12, † 16)	no	no	no	10–15%	frontal	left	yes	2
**IPDT 06** (⟳ 4, † 9)	no	no	no	20%	temporal	right	yes	0
**IPDT 07** (110)	yes	no	no	25%	temporal	left	yes	0
**IPDT 08** (⟳ 61, † 66)	no	no	no	25%	median frontal gyrus	left	yes	3
**IPDT 09** (114)	partially	yes	no	85%	temporal	left	yes	4
**IPDT 10** (⟳ 16, † 28)	no	no	no	30%	central gyrus and subcentral lobe	left	yes	0
**IPDT 11** (⟳ 6, † 9)	no	no	no	21%	superficial parietal gyrus	left	no	0
**IPDT 12** (⟳ 7, † 8)	yes	no	no	7%	parieto-occipital	left	yes	2
**IPDT 13** (⟳ 10, † 25)	partially	no	no	15%	temporoparietal	left	partially	4
**IPDT 14** (⟳ 36, † 44)	yes	no	no	15%	parieto-occipital	left	partially	8
**IPDT 15** (⟳ 7, † 9)	no	no	no	10%	temporal/parietal	left	n/a	n/a
**IPDT 16** (⟳ 18, † 36)	yes	no	no	28%	parietal	left	yes	5

PFS = ⟳ = progression-free survival, OS = † = overall survival, n/a = data not available, and ^b^ = grouping: MGMT unmethylated vs. MGMT methylated (partially and fully).

**Table 3 cancers-15-02603-t003:** Survival times, OS, PFS, and PPS (median, [95% CI]), obtained for the iPDT cohort in this study, overall and split according to MGMT promoter methylation, in comparison to survival times reported for other therapeutic methods in the literature.

		iPDT All Cases	iPDT MGMT Unmethylated	iPDT MGMT Methylated	Stupp [2] ^c^	Chemoradiation [10] ^d^	CyberKnife^®^ [12] ^d^	Stereotactic Radiosurgery [11] ^c^
Subjects (*n*)		16	8	8	287	56	12	30
PFS (months)	Median [95% CI]	16.4 [5.1; 27.6]	8.3 [1.9; 14.7]	35.7 [0.0; 100.7]	6.9 [5.8; 8.2]	8.0 [5.6; 10.4]	16.0 [10.4; 21.6]	8.2 [4.6; 10.5]
OS (months)	Median [95% CI]	28.0 [6.0; 50.0]	15.0 [5.3; 24.7]	43.9 [0.0; 135.4]	14.6 [13.2; 16.8]	12.0 [9.6; 14.4]	18.0 [10.9; 25.1]	14.8 [10.9; 19.9]
PPS (months)	Median [95% CI]	6.3 [1.6; 11.0]	3.8 [1.3; 6.4]	15.3 [0.0; 31.4]	n/a	4.0 [3.0; 5.0]	3.0 [2.0; 4.0]	n/a

PFS = progression-free survival, OS = overall survival, PPS = postprogression survival, CI = confidence interval, n/a = data not available, **^c^** = values from publication, and **^d^** = values calculated.

**Table 4 cancers-15-02603-t004:** *p*-values according to pairwise log-rank testing (SPSS) of differences among some of the survival times listed in Table 3. Significant differences are highlighted with **boldface** formatting.

	iPDT MGMT Methylated vs. Unmethylated	iPDT vs. Stupp [2]	iPDT vs. Chemoradiation [10]	iPDT vs. CyberKnife^®^ [12]	iPDT vs. Stereotactic Radiosurgery [11]
PFS	**0.030**	**<0.001**	**0.005**	0.341	n/a
OS	**0.031**	**0.017**	**0.022**	0.280	**0.036**
PPS	0.192	n/a	0.516	0.237	n/a

PFS = progression-free survival, OS = overall survival, and PPS = postprogression survival.

**Table 5 cancers-15-02603-t005:** Course of different tumor-related volumes of interest for the iPDT-treated patient cohort (values rounded to full mm²).

Volume	Tumor Pre iPDT (mm³)	CE Pre iPDT (mm³)	Necrosis Pre iPDT (mm³)	CE 1 Day Post iPDT (mm³)	iPDT Remnant 1 Day Post iPDT (mm³)	iPDT Remnant 3 Months Post iPDT (mm³)	iPDT Remnant 3 Months Pre Recurrence (mm³)	iPDT Remnant at Recurrence (mm³)	Recurrent Tumor at Recurrence (mm³)
**Median**	6063	3533	1350	535	12,426	5585	4036	3914	5893
**Average**	7622	5180	2442	1108	19,166	7151	4474	4556	6645
**Max**	21,825	12,362	9463	4720	66,857	20,027	12,254	9607	17,177
**Min**	1362	878	118	0	2062	370	305	157	6
**SD**	5849	3761	2579	1463	16,571	5395	3188	2917	6115

CE = contrast enhancement; SD = standard deviation.

**Table 6 cancers-15-02603-t006:** Median survival values obtained separately for unmethylated and methylated MGMT promoters for iPDT in this study and for sole chemoradiation treatment [56] (values rounded).

	iPDT		Chemoradiation [56]	
	MGMT Unmethylated	MGMT Methylated	MGMT Unmethylated ^e^	MGMT Methylated ^e^
PFS median (months)	8.3	35.7	5.3	10.3
OS median (months)	15.0	43.9	12.7	21.7

**PFS** = progression-free survival, **OS** = overall survival, and **^e^** = values from publication.

## Data Availability

No new data were created or analyzed in this study. Data sharing is not applicable to this article.

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
