# Peer review of "Interstitial Photodynamic Therapy of Glioblastomas: A Long-Term Follow-up Analysis of Survival and Volumetric MRI Data"

_cancers, 2023, doi:10.3390/cancers15092603_

Round 1

Reviewer 1 Report

The novelty in this paper appears to be in the use of MRI parameters as predictors of survival. With 16 glioblastoma patients treated with PDT, of which 11 ones were already reported in ref.21, the study is small. Unfortunately, the MRI assessments are limited to volumetry for necrosis and contrast enhancement, and do not include quantification of MRI signals or novel MRI methods such as diffusion weighted imaging. This probably explains why results are negative compared with those reported by Fujita et al. in 30 patients [J Neuro-Oncol 2021;155:81-92; not cited]. The inclusion of quantitative data e.g. the extent of contrast enhancement might increase the significance of the results.

Reviewer 2 Report

In the present work, the authors analyze the effect of iPDT in Glioblastoma as compared to other treatment strategies. The work is quite significant and can be accepted in present form.   

Author Response

Point 1: In the present work, the authors analyze the effect of iPDT in Glioblastoma as compared to other treatment strategies. The work is quite significant and can be accepted in present form.

Response 1: We thank you very much for reviewing our manuscript and for your supportive statement.

Reviewer 3 Report

Glioblastomas (GBM) are considered one of the most aggressive tumors and those with the worst prognosis after diagnosis of the central nervous system. Standard treatment for this neoplasm includes surgical resection of the primary tumor followed by a schedule of radiation and chemotherapy with Temozolomide (Stupp protocol) with a survival of 12-15 months after treatment. The latest WHO classification of CNS tumors classified GBMs as grade 4 adult-type diffuse gliomas with wild-type IDH accompanied by microvascular proliferation or necrosis, or TERT promoter mutations or EGFR amplification, or chromosome 7 gain/loss. The activity of the enzyme O6-methyl guanine-DNA methyltransferase (MGMT), which is responsible for the repair of guanine alkylated by TMZ, has been reported to be associated with patients' resistance to TMZ therapy, while patients with silenced MGMT showed better survival, with an OS of 24.59 months compared with 14.11 months observed in patients with unmethylated MGMT promoter [56]. Determination of MGMT promoter methylation is an excellent prognostic marker, and about 40% of GBMs have methylation at the promoter. In addition, the presence of TMZ-resistant glioma stem cells has been associated with failure of therapy.

In recent years, Tumor Treating Fields (TTFs) have gained attention as a noninvasive treatment for patients diagnosed with GBM and involve the generation of a biophysical force on dipoles through the delivery of low-intensity, intermediate-frequency electric fields. TTFs interfere with cancer cell proliferation by disrupting cell division and inducing cell death [60]. In a randomized trial, the combination of TTFs with TMZ treatment improved the OS and PFS of patients compared with those who received only the maintenance scheme with TMZ.

The U.S. Food and Drug Administration recently approved 5-aminolevulinic acid (ALA) for fluorescence-guided surgery (FGS) of tumors. As a result, interest in using this precursor of PS to administer PDT has been renewed, and PDT is becoming an alternative approach to the treatment of several diseases, including cancers. The study of PDT for the treatment of GBM has gained attention in recent years, where the most widely used photosensitizers for PDT of GBM have been reported to be porphyrins, chlorines, and phthalocyanines, and also their precursors, such as in the case of aminolaevulinic acid, in the past five years.

Second-generation PSs have been developed to reduce the drawbacks associated with first-generation PSs. These PSs are chemically pure and have a maximum absorption in the phototherapeutic window (600-850 nm), a higher O2 formation yield, and a higher molar extinction coefficient. Compounds classified as second-generation PSs include porphyrin derivatives, 5-aminolevulinic acid, chlorines, and phthalocyanines. Most second-generation PSs are lipophilic and are not soluble in an aqueous environment, which greatly hinders intravenous administration of PSs. In addition, these PSs tend to aggregate, compromising their photochemical properties and bioavailability at the active site. To improve solubility in aqueous media, second-generation PSs have been incorporated into different nanocarriers with or without conjugation with active targeting agents, making third-generation PSs. However, the reach and accumulation of large molecules (e.g., siRNA, mRNA, antibodies, etc.) in the brain is significantly hindered by BBB, and the activity of efflux transporters expressed in endothelial cells pumps back small lipophilic molecules that achieve passive diffusion across the plasma membrane to the bloodstream, making it difficult to achieve optimal therapeutic concentrationsIn this case, PDT is a good solution because of its ability to induce vascular damage, sometimes triggering vessel collapse and consequently disrupting the integrity of the BBB. PDT triggers a reversible BBB permeabilization associated with a dose-dependent decrease in tight junctions in the vascular endothelium thus providing promising features as a tool for the BBB opening and drug brain delivery.          

CONS of PDT use

Besides the efforts continuously made by several research groups worldwide, GBM is considered one of the most treatment-resistant tumors. However, depending on the therapeutic strategy adopted, PDT is considered a promising therapeutic strategy for treating GBM patients, but as for other therapeutic approaches, GBM developed several mechanisms of resistance that impair its success in eliminating the tumor completely. As one of the main limitations of PDT is the arising of tumor cell resistance

Among mechanism involved in GBM PDT induced resistance:  NO levels that are enhanced by PDT treatments and that  are responsible of  inhibition of tumor suppression mechanisms. An enhancement in the DNA oxidative damage repair system and the mechanism of DNA break repair have been observed after PDT treatments. Hypoxia and angiogenesis: most of the current PS used in PDT generated ROS through a type-II mechanism that is dependent on molecular oxygen, leading to severe tumor hypoxia by either consuming the oxygen by the PS reaction or via vascular collapse eventually enhancing the proliferation of CD133-positive glioma stem cells (GSCs). Moreover,  tumor regrowth in newly treated GBM patients is mainly attributed to stem cell survivla under hypoxic conditions. In addition, PDT-mediated hypoxia triggers molecular mechanisms associated with cell survival, including the release of pro-angiogenic associated growth factors, for example, VEGF, epidermal growth factor (EGF), and angiopoietin (ANGPT) which improve proliferation, migration and invasion. Finally, the redox status of the cells allow cell detoxification from reactive species generated by PDT, thus enhancing GSCs to induce tumor progression and resistance.

The paper is quite interesting, dealing with one of the most promising cancer treatment techniques. It has an important criticality due to the low number of patients treated with iPDT compared with Stupp's larger cohort as reported in Table 3. However, the study is well conducted and the experimental design is clear.

Major revision

With reference to the referee's comments, resistance to induced PDT is a relevant topic that needs to be better addressed by the authors.

Round 2

Reviewer 1 Report

Acceptable revision

Reviewer 3 Report

The paper can be accepted for publication in its current version